# Differences in macular capillary parameters between healthy black and white subjects with Optical Coherence Tomography Angiography (OCTA)

**Lindsay Y. Chun**[1,2], **Megan R. Silas**[2], **Rose C. Dimitroyannis**[2,3], **Kimberly Ho**[2,3], **Dimitra Skondra**[2,4]*

**1** University of Chicago Pritzker School of Medicine, Chicago, IL, United States of America, **2** Department of Ophthalmology and Visual Science, University of Chicago Medical Center, Chicago, IL, United States of America, **3** University of Chicago, Chicago, IL, United States of America, **4** J. Terry Ernest Ocular Imaging Center, University of Chicago Medical Center, Chicago, IL, United States of America

* Dskondra@bsd.uchicago.edu

## Abstract

### Purpose

To investigate if there are differences in macular capillaries between black and white subjects using optical coherence tomography angiography (OCTA) and identify potential factors underlying the epidemiologically-based higher vulnerability of black populations to diabetic retinopathy (DR).

### Methods

This prospective, observational cross-sectional study included 93 eyes of 47 healthy subjects with no medical history and ocular history who self-identified as black or white and were matched for age, sex, refractive error, and image quality. Subjects underwent OCTA imaging (RTVue-XR Avanti) of the superficial (SCP) and deep (DCP) capillary plexuses and choriocapillaris. AngioAnalytics was used to analyze vessel density (VD) and choriocapillaris % blood flow area (BFA) in the 1mm-diameter fovea, parafovea, and 3mm-diameter circular area including the fovea and parafovea (3x3mm image). Foveal avascular zone (FAZ) was also analyzed. Linear mixed models were used to evaluate for differences between the study groups.

### Results

Compared to the white subjects in this study, black subjects were found to have: lower foveal VD in the SCP (p<0.05); lower VD in the parafovea and in the 3x3mm image in the DCP (p<0.05); larger FAZ in SCP and DCP (p<0.05); and decreased choriocapillary BFA in the area underlying the fovea, parafovea, and 3x3mm image (p<0.05).

**Data Availability Statement:** All relevant data are within the manuscript and its Supporting Information files.

**Funding:** LC; No grant number; University of Chicago Pritzker School of Medicine (Pritzker Research Fellowship); https://pritzker.uchicago.edu/page/pritzker-and-northshore-fellowships; The funders had no role in study design, data collection and analysis, decision to publish, or preparation of the manuscript. DS; No grant number; Illinois Society for the Prevention of Blindness (Research Grant); http://www.eyehealthillinois.org/research-grants/; The funders had no role in study design, data collection and analysis, decision to publish, or preparation of the manuscript.

**Competing interests:** The authors have declared that no competing interests exist.

## Conclusion

In our study, our black subjects had decreased macular capillary vasculature compared to matched white subjects, even in early adulthood and the absence of any systemic or ocular conditions. To our knowledge, this is the first report showing that retinal and choriocapillary vascular differences may contribute to racial disparities in vulnerability to DR.

## Introduction

Diabetes mellitus (DM) is a serious epidemic disease in the United States, with 9.4% of the population afflicted with the chronic disease [1]. DM disproportionately affects 13.4% of black adults compared to 7.3% of non-Latino white adults in the US [2]. Black populations also face a larger risk of developing debilitating complications from DM, including diabetic retinopathy (DR), kidney failure, and lower-extremity amputation [2]. DR, the leading cause of blindness in working-age adults, has higher prevalence in black populations than in white populations with DM [3–17]. Black individuals also tend to have more severe and vision-threatening forms of DR compared to white individuals [3,4]. However, studies that have attempted to identify the medical and genetic factors underlying the higher vulnerability to DR in black populations are limited and have had conflicting results [4,5,12,14,15,18,19]. While these factors may have varying effects on the total risk of developing DR, the retinal vasculature has not yet been directly studied in different populations to further investigate the existing race-based disparities of DR.

Previous studies have investigated differences in retinal anatomy by race. One group found that black adults had smaller depth and larger diameter of the foveal pit compared to their white counterparts [20]. Although some data indicate that black individuals have thinner retinas at the fovea than white individuals, it has been suggested that the morphology of the foveal pit can modify the true measurement of retinal thickness, and that previous thickness measurements may not reflect a true difference [20,21]. These studies suggest that structural features of the fovea may play a role in increasing the susceptibility to retinal disease, but functional studies of the retinal microvasculature in a wider diversity of racial and ethnic populations have not yet been performed to elucidate the mechanisms of this differential susceptibility.

Optical coherence tomography angiography (OCTA) is a novel, non-invasive imaging technology that provides high-resolution imaging of the *in-vivo* retinal vasculature within seconds. By rendering *en face* images of different retinal vascular layers, OCTA produces 3-dimensional visualization and quantitative characteristics of vascular structures, such as vessel density (VD) per retinal vascular layer, % blood flow area (BFA), and area of foveal avascular zone (FAZ), which traditional imaging techniques cannot provide [22]. OCTA has been used to study the chorioretinal vasculature in healthy populations [23–30] and in states of ocular disease [31–40]. Several studies using OCTA have reported that there is decreased density of retinal capillaries in diabetics without DR have compared to controls, and decreased density with the progression of DR [36, 37, 41–43]. However, these studies either did not include black subjects or did not differentiate their findings by race or ethnicity. Given the epidemiologically evident vulnerability of black populations to DR, the vascular characteristics of the retina among different demographic groups in a disease-free state require further examination.

In this study, we investigated the retinal and choriocapillary vasculature of young healthy adult black and white subjects with OCTA to capture subjects' VD in the superficial and deep

capillary plexuses (SCP and DCP, respectively); BFA in the choriocapillaris; and FAZ in the SCP and DCP and observed for differences between the groups.

## Design and methods

This prospective cross-sectional study of healthy participants was approved by the Institutional Review Board of the University of Chicago (IRB #18–117, #17–0715). All study protocols adhered to the tenets of the Declaration of Helskinki [44]. The study conformed to the Health Insurance Portability and Accountability Act of 1996 regulations. The study was conducted between September 2017 and February 2018. All subjects provided informed, written consent.

### Participants

Young, healthy adult subjects (≥18 years old) without any history of any medical or ocular diseases, and unremarkable findings on OCTA imaging were included in this study. Subjects were clinic patients, students, and other members of the university and surrounding local community who were either approached and informed by study staff of this study, or who contacted study staff to participate after learning of the study independently via word-of-mouth during the time interval of this study. All analyzed subjects were assigned a randomly-generated number and provided demographic information including medical, ocular, and social history (S1 Table). Subjects self-identified their racial or ethnic identities by being asked to select from a list that included: black, white, East Asian, Southeast Asian, Indian, Native American, Latino, mixed, and other. Subjects who selected categories that were not solely "black" or "white," or who selected multiple categories, were excluded from analysis due to low number in those group (fewer than 10 subjects per non-black and non-white groups). Subjects were also asked if they currently or had ever smoked or used tobacco products. Subjects who had never smoked or used tobacco products, or smoker fewer than 100 cigarettes in their life and currently did not smoke or use other tobacco products at the time of imaging, were categorized as not being a regular smoker and included in the study; others were excluded from this study (S1 Table) [45]. Subjects who had a history of systemic medical conditions for which they required medications or interventions (such as diabetes, hypertension, hyperlipidemia, invasive or metastatic cancer, pulmonary conditions, coagulopathies) or ocular conditions (such as glaucoma, retinal disorders, uveitis, cataracts, blindness), and other conditions that could potentially affect the retinal microvasculature or prevent adequate imaging with OCTA, were excluded from this study. We determined that matching the subjects for blood pressure and other biomarkers of systemic health was not necessary, as we studied adult subjects in states of optimum health at baseline and prior to the onset of systemic conditions. Chart review was performed to confirm subjects' available history. Subjects' spherical equivalent (SEq, a sum of spherical power + half of cylinder power) were calculated by using their refractive errors, which were obtained through autorefraction (KR-8000 AutoKerato Refractometer, Topcon Omni Systems, Oakland, NJ, USA), lensometry (Topcon CL-2000 Computerized Lensmeter, Topcon Omni Systems, Oakland, NJ, USA), or copies of their most recent glasses prescriptions. Refractive data of each eye included in analysis can be found in S1 Table.

### OCTA Imaging

Images were obtained using the Optovue RTVue XR Avanti (Optovue Inc, Fremont, California, USA Version 2016.2.0.35) with the AngioRetina mode (3x3mm macular cube). Each image was made up of 304 clusters of repeated B-scans containing 304 A-scans, and the images were automatically segmented. The software set the SCP 3um beneath the inner limiting membrane (ILM) and 15um beneath the IPL; the DCP was set between 15um and 70um beneath

the IPL, and the choriocapillaris was set between 30um and 60um beneath the retinal pigment epithelium (RPE). Each image was assessed for quality by a reader (LC) masked to quantitative results of OCTA. The threshold for signal strength intensity (SSI) was set at $\geq 50$ based on previous studies [27,46]. OCTA data of each eye included in our analysis can be found in S1 Table.

### Analysis of VD and FAZ at the SCP and DCP

Vessel density (VD) measurements were associated with OCT images segmented at the corresponding level (Fig 1A and 1B). To generate VD, the software automatically overlaid an annulus on the 3x3mm macular scan centered on the fovea. The annulus was composed of 2 concentric circles of 1mm and 3mm diameters. The fovea was within the inner circle, and the parafovea was defined as the ring between the inner and outer circles. VD represents the percentage of pixels in the image corresponding to blood vessels and was automatically calculated with the AngioAnalytics software. The parafovea was divided automatically into superior and inferior hemifields and superior, temporal, inferior, and nasal quadrants.

The FAZ (in mm$^2$) at the level of the SCP and DCP was manually measured with built-in software (Fig 1C and 1D). The center of the FAZ was selected and the built-in nonflow area measurement tool automatically calculated the area of FAZ [30].

### Analysis of BFA of the choriocapillaris

BFA is the percentage of the retinal image occupied by pixels corresponding to blood vessels and is automatically calculated after the image capturing process with AngioAnalytics (Fig 1E–1G). Previous studies have found that BFA and VD may be considered related variables [33, 37]. This measure was obtained for the choriocapillaris. BFA was determined for the circular areas of 1mm and 3mm diameter manually centered in the area below the fovea. BFA for the 3x3 macular image was determined from the values for the circular area of 3mm diameter centered at the fovea (Fig 1E). BFA was also determined in the area below the parafovea by subtracting the BFA of the 1mm-diameter circle (Fig 1F) from the BFA of the 3mm-diameter circle (Fig 1G) in the choriocapillaris, resembling an annulus centered at the fovea similar to that automatically rendered for VD calculation. A similar technique for measuring blood flow in the choriocapillaris has previously been described using the entire 3x3mm macular image [33].

### Statistical analysis

Statistical analysis was performed using Stata13 (College Station, TX: StataCorp LP). Nonparametric Wilcoxon rank-sum (Mann-Whitney) tests were performed to compare continuous demographic and technology-related variables between the two groups. These data are presented as means ± standard deviation. Linear mixed models were used to estimate the differences in retinal characteristics between racial groups. Linear mixed models were used to account for potential correlations between the left and right eyes of an individual. All retinal data are presented as means ± standard deviation unless specified otherwise. A p-value of $<0.05$ was considered statistically significant.

## Results

93 eyes of 47 mostly young, healthy, nonsmoking adult subjects were included in this study (Table 1). 23 subjects self-identified as black (age range: 22.4–32.4y, n = 21; 41.1–47.5y, n = 2) and 24 as white (age range: 22.3–32.4y, n = 21, 40.3–48.0, n = 3) (both groups non-Latino).

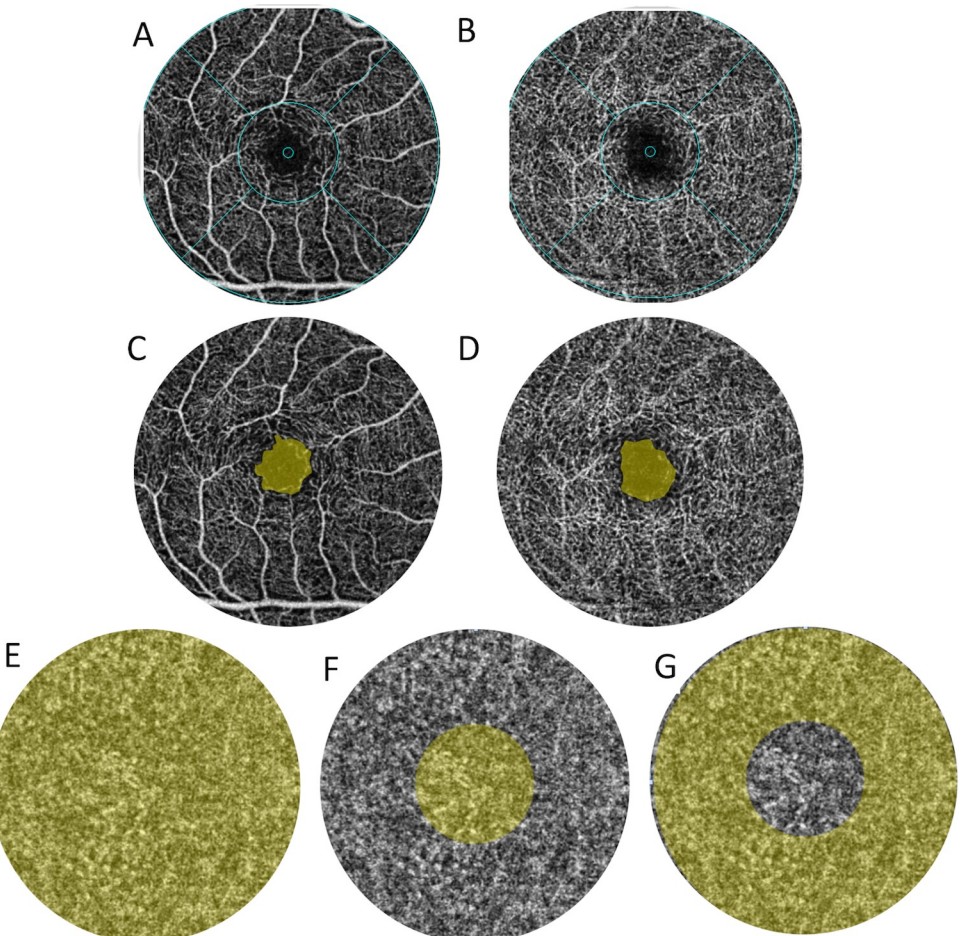

**Fig 1. Methods for calculating VD, BFA, and FAZ from *en face* OCTA images.** All images are from the same eye of a single subject. (A, B) *En face* OCTA images representing the VD calculated with the annulus automatically centered at the level of the SCP (A) and DCP (B). (C, D) Measurement of FAZ in the SCP (C) and DCP (D) using built-in software with OCTA AngioAnalytics. (E-G) *En face* OCTA images representing the areas (shaded in yellow) selected to measure the BFA in the choriocapillaris in the region below the 3x3mm macular area (E), central 1mm circular area of the fovea (F), and parafovea (G). VD = vessel density (%), SCP = superficial capillary plexus, DCP = deep capillary plexus, BFA = blood flow area (%), FAZ = foveal avascular zone ($mm^2$).

Medical and ocular history were obtained through subject interview. All subjects were non-smokers. The distribution of genders was similar in both groups (p = 0.475). There were no significant differences in age, signal strength intensity (SSI), and spherical equivalent between

**Table 1. Characteristics of the subject groups.**

|  | Black | White | p |
|---|---|---|---|
| # Subjects (Eyes) | 23 (46) | 24 (47) | |
| Gender | 11 f; 12 m | 14 f; 10 m | 0.48 |
| Age (y) | 27.86 ± 5.87 | 29.79 ± 6.08 | 0.09 |
| OU SSI | 70.93 ± 7.93 | 71.13 ± 7.23 | 0.96 |
| Ou SEq | -2.86 ± 2.71 | -2.48 ± 2.50 | 0.41 |

Data are presented as means ± standard deviation. f = female, m = male, OU = both eyes, SSI = signal strength intensity, SEq = spherical equivalent. Wilcoxon (Mann-Whitney) rank-sum test used to compare the study groups by race.

**Table 2. VD, BFA, and FAZ in the SCP, DCP, and choriocapillaris for black and white subject groups.**

| | Black | White | p |
|---|---|---|---|
| **SCP VD (%)** | | | |
| Macula | 52.37 ± 3.59 | 53.40 ± 2.79 | 0.177 |
| Fovea | 28.62 ± 5.44 | 32.21 ± 4.08 | 0.011* |
| Parafovea | 55.01 ± 3.85 | 55.70 ± 3.09 | 0.407 |
| Superior hemifield | 55.37 ± 3.66 | 55.79 ± 3.12 | 0.590 |
| Inferior hemifield | 54.66 ± 4.15 | 55.62 ± 3.27 | 0.292 |
| Temporal | 54.10 ± 3.62 | 54.57 ± 3.15 | 0.565 |
| Superior | 56.17 ± 4.12 | 56.79 ± 3.23 | 0.453 |
| Nasal | 54.60 ± 3.79 | 55.30 ± 3.18 | 0.382 |
| Inferior | 55.20 ± 4.48 | 56.18 ± 3.68 | 0.326 |
| **SCP FAZ (mm2)** | | | |
| | 0.30 ± 0.11 | 0.22 ± 0.07 | 0.006* |
| | Black | White | p |
| **DCP VD (%)** | | | |
| Macula | 59.70 ± 2.37 | 61.08 ± 1.71 | 0.004* |
| Fovea | 29.84 ± 6.97 | 32.98 ± 5.01 | 0.079 |
| Parafovea | 62.64 ± 2.54 | 63.83 ± 1.89 | 0.025* |
| Superior hemifield | 62.76 ± 2.35 | 63.82 ± 2.15 | 0.033* |
| Inferior hemifield | 62.52 ± 2.90 | 63.84 ± 1.91 | 0.031* |
| Temporal | 61.81 ± 2.62 | 62.73 ± 2.22 | 0.106 |
| Superior | 63.73 ± 2.68 | 64.97 ± 2.18 | 0.018* |
| Nasal | 62.03 ± 2.64 | 62.27 ± 2.07 | 0.025* |
| Inferior | 63.00 ± 3.10 | 64.35 ± 2.13 | 0.038* |
| **DCP FAZ (mm2)** | | | |
| | 0.34 ± 0.11 | 0.26 ± 0.08 | 0.008* |
| | Black | White | p |
| **Choriocapillaris % Blood flow area in 3x3mm image** | | | |
| | 61.00 ± 1.77 | 62.70 ± 1.27 | <0.001* |
| **Choriocapillaris % Blood flow area at fovea** | | | |
| | 60.64 ± 2.81 | 62.45 ± 2.30 | 0.003* |
| **Choriocapillaris % Blood flow area at parafovea** | | | |
| | 61.04 ± 1.69 | 62.73 ± 1.21 | <0.001* |

Data for the macula refer to the circular area including parafovea and fovea in the 3x3mm image. Data are presented as means ± standard deviation. VD = vessel density (%), BFA = blood flow area (%), FAZ = foveal avascular zone (mm$^2$), SCP = superficial capillary plexus, DCP = deep capillary plexus.
*p<0.05, linear mixed models.

black and white groups (p = 0.09, 0.96, 0.41, respectively). 2 subjects with a history of prior refractive surgery (LASIK) provided their pre-surgical refractive errors. Seven eyes in each group had high myopia (≥6.00 diopters).

The values of the VD, BFA, and FAZ are represented in Table 2 and Fig 2. Since the fovea is mostly avascular, any foveal differences may be best represented by the FAZ. However, due to the fact that foveal VD data is automatically used by the OCTA software to calculate the VD in the 3x3mm circular macula, we have included the foveal VD data. In the SCP (Fig 2A–2D), black subjects had significantly lower foveal VD than white subjects (28.62 ± 5.44% vs 32.21 ± 4.08%, p = 0.011). In the SCP parafoveal VD, there was no significant difference in the VD between black and white subjects (55.01 ± 3.85% vs 55.70 ± 3.09%, p = 0.407). When the

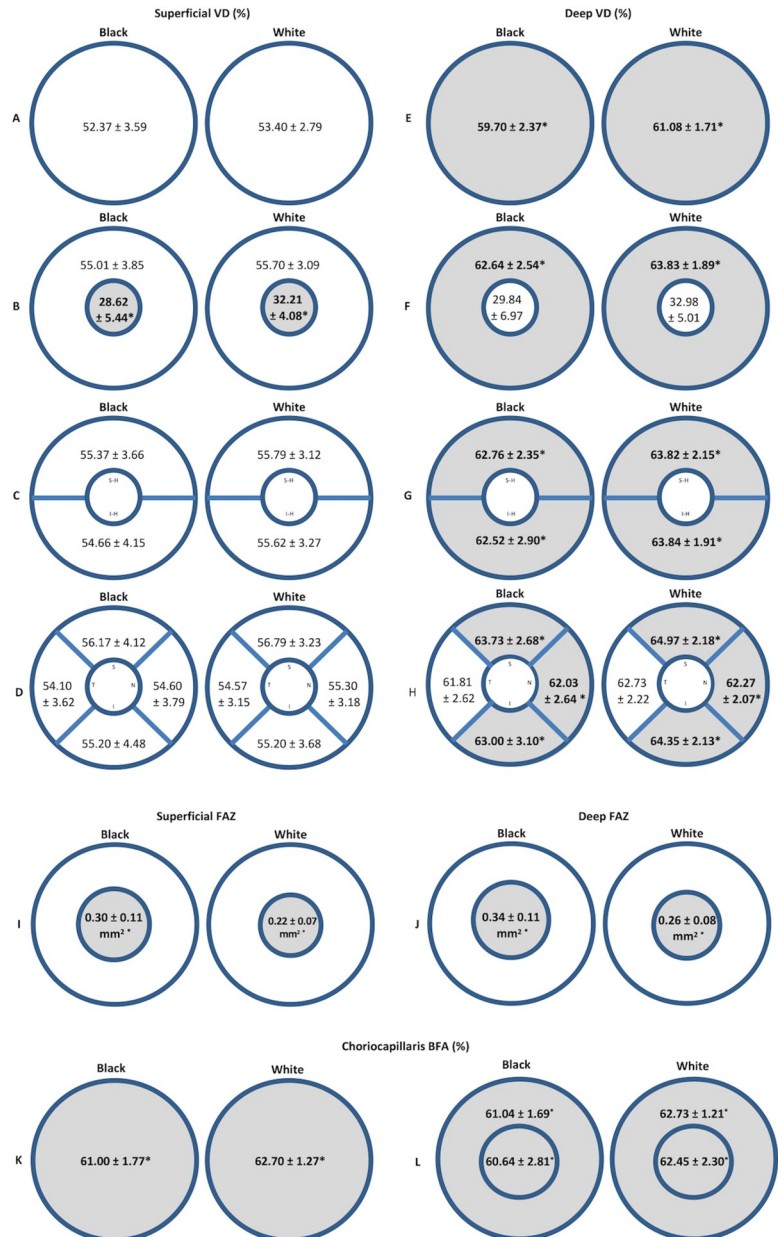

**Fig 2. Representative maps of mean VD (A-D) and FAZ (E) in the SCP and DCP, and BFA (F) in the choriocapillaris for both subject groups.** Data are presented as means ± standard deviation. Portions of the map shaded in gray represent areas where there was a statistically significant difference between the two subject groups. Maps representing the VD in the 3x3mm area of the macular image (A), parafovea and fovea (B), superior and inferior hemifields (C), and parafoveal quadrants (D). Maps representing the area of FAZ in the SCP and DCP (E). VD = vessel density (%), FAZ = foveal avascular zone (mm$^2$), BFA = blood flow area (%), SCP = superficial capillary plexus, DCP = deep capillary plexus, S-H = superior hemifield, I-H = inferior hemifield, S = superior, N = nasal, I = inferior, T = temporal. *p<0.05, linear mixed models. See Table 1 for corresponding p values.

parafovea was divided into quadrants and superior and inferior hemifields, there was no significant difference between the study groups (p = 0.590, p = 0.292, respectively). In the region of the macula analyzed in the 3x3mm scan including both fovea and parafovea, there was no significant difference in the VD (52.37 ± 3.59% vs 53.40 ± 2.79%, p = 0.177).

Black subjects had significantly larger FAZ than white subjects in the SCP ($0.30 \pm 0.11 mm^2$ vs $0.22 \pm 0.07 mm^2$, p = 0.006; Fig 2I).

In the DCP (Fig 2E–2H), black subjects had lower parafoveal VD ($62.64 \pm 2.54\%$ vs $63.83 \pm 1.89\%$, p = 0.025), and this difference persisted in the superior and inferior hemifields and all quadrants except in the temporal quadrant compared to white subjects. There was a trend towards lower foveal VD in black subjects, but this did not reach statistical significance ($29.84 \pm 6.97\%$ vs $32.98 \pm 5.01\%$, p = 0.079). In the region of the macula analyzed in the 3x3mm image including both fovea and parafovea, black subjects had lower VD ($59.70 \pm 2.37\%$ vs $61.08 \pm 1.71\%$, p = 0.004).

Black subjects had significantly larger FAZ in the DCP compared to white subjects ($0.34 \pm 0.11 mm^2$ vs $0.26 \pm 0.08 mm^2$, p = 0.008; Fig 2J).

Black subjects had significantly lower BFA (Fig 2K and 2L) in the choriocapillaris compared to white subjects in the circular 3x3mm macular image ($61.00 \pm 1.77\%$ vs $62.70 \pm 1.27\%$, p<0.001), in the central 1mm-diameter circular area at the position of the fovea ($60.64 \pm 2.81\%$ vs $62.45 \pm 2.30\%$, p = 0.003), and in the annulus-shaped area of the parafovea ($61.04 \pm 1.69\%$ vs $62.73 \pm 1.21\%$, p<0.001).

## Discussion

There is currently a deficit in the knowledge of the factors that make black populations more vulnerable to developing DR and other diabetic microvascular complications. Our study is the first, to our knowledge, to investigate the unresolved racial disparities in ocular health with OCTA and uncover possible baseline differences of the retinal vasculature between young, healthy adult black and white subjects. Our data show that black subjects had lower macular VD in the DCP, lower macular VD at the fovea in the SCP, larger FAZ in both the SCP and DCP, and lower choriocapillary BFA compared to white subjects (Table 2, Fig 2). Our data suggest that anatomically-related elements within the pathophysiological sequences of micro-vascular retinal diseases may contribute to the dissimilar susceptibility of people of different racial backgrounds.

National US data show that there is a higher prevalence of DM and DR among black individuals compared to their white counterparts [1,19]. Although racial and ethnic identity are often employed as categorical (albeit imperfect) social factors in epidemiological studies, there are conflicting data regarding the underlying etiology of the risk of DR in different racial groups. Some studies have suggested that higher DR prevalence in black populations is related to DM risk factors and severity (including age, gender, duration of DM, serum glucose, hemoglobin A1C, and use of anti-DM medications) [12,14,18]. Other studies, however, found that black subjects had a higher risk of DR compared to white subjects even after adjusting for DM risk factors [4,5,14,15]. Thus, while there are consistent race-based differences in the reported prevalence of DR and vision loss, there are conflicting data on the underlying etiology of the observed discrepancies.

Other investigations have found that socioeconomic disparities affect vision-related health among different racial and ethnic groups in the US, but these findings do not explain why there are racial differences in DR prevalence even when controlling for factors related to DM severity [47,48]. Thus, while there are consistent race-based differences in the prevalence of DR, there are conflicting epidemiologic data on the comparative race-based incidence of DR. Our findings support our hypothesis that there may be baseline differences in the retinal vasculature between our black and white subjects as seen with OCTA that may contribute to the differences in the risk of developing DR.

Previous studies have attempted to determine the role of genetics to elucidate the racial discrepancies in DR prevalence, but have been unable to identify high-risk alleles associated with the development of DR in black populations [17,49–52]. Although the identification of a set of candidate genes with robust association with DR could have a major role in the risk of disease development, the moderate role of heritability and genetics in DR among black populations suggests that other factors hold an important function in its pathophysiological sequence.

Lower capillary density in the DCP found in our study could potentially have some physiological consequences in normal states. A study of oxygen distribution through the retinal layers in rat models identified the photoreceptor inner segment and the inner and outer plexiform layer (IPL and OPL, respectively) as the retinal layers with the largest oxygen requirements and high sensitivity to changes in vascular supply [53,54]. These regions have the highest metabolic demands because these are sites of intercellular communication between ganglion and bipolar cells (in the IPL) and the inner segments of photoreceptors and horizontal cells (in the OPL) [53–55]. In the OCTA segmentation scheme used in our study, the DCP supplies the outer elements of the IPL and OPL [54,55]. Therefore, our data demonstrate that there may be a slightly decreased vascular supply from the DCP to the most metabolically active retinal layers in black subjects compared to white subjects, even in the absence of disease.

OCTA studies have implicated the DCP as a key location of the earliest evidence of parafoveal capillary loss and nonperfusion in DR. Different studies found significantly decreased VD in the DCP of DM patients with no or mild DR compared to non-DM controls, and surmised that the vulnerability of the DCP in the setting of DR could be due to its location in a watershed zone of blood supply [36,37,41,43,45]. The key site of ischemic injury has been localized to the DCP in studies of retinal artery occlusions, diabetic macular edema (DME), and with increasing severity of DR [22,35,39,40,56]. These studies highlight the importance of the DCP because capillary loss in this layer is an early sign of retinal ischemic disease. However, these studies were either carried out largely in nonblack subjects or did not describe their subjects' race or ethnicity.

In light of these previous studies, it could be hypothesized that compared to our white subjects, our black subjects had lower VD in the regions of the retina that are the most metabolically active and most sensitive to ischemic insults. Reduced vasculature in the DCP may lead to higher risk for diabetic ischemic injury, because the DCP supplies the most metabolically active and oxygen-dependent region of the retina. We suggest that baseline vascular differences in the DCP could contribute, along with other risk factors, to the earlier development and higher prevalence of DR. As individuals age and sustain chronic damage to the integrity of their microvasculature due to systemic conditions like hypertension and DM, those who have slightly fewer capillary blood flow at baseline could theoretically be more vulnerable to microvascular injury and compensate for damage—especially if the chronic conditions are not well-controlled. Based on our findings, variances in susceptibilities to other vasculopathies of the retina, such as lupus and sickle cell anemia, could also be based in functional differences in retinal vascularization. Further studies are needed to investigate these relationships.

In our study, BFA of the choriocapillaris was also lower in black subjects compared to white subjects. The choriocapillaris is a key supplier of oxygen to the outer segments of the photoreceptors, while the vessels of the DCP supply the synapses between the inner segments of photoreceptors and horizontal cells [54,55]. Studies have suggested that the choriocapillary circulation is compromised in the setting of DM with and without and DR [37,57]. Whether derangements of capillary function of the choriocapillaris or neuroretinal degeneration of the photoreceptor occur first remains undetermined [58,59]. Similar to the potential role of decreased intraretinal VD, the lower BFA in the choriocapillaris of black subjects may suggest that they have fewer vessels to compensate for choroid capillary dropout in the setting of DM.

However, the quantification of BFA in the choriocapillaris may have been affected by the larger amount of light absorption by the higher melanin content in the choroid of black subjects, so we cannot be certain that decreased choriocapillaris BFA seen in black subjects is representative of a true difference in the measured flow and not the result of artifacts related to different pigment content and light absorption [60,61]. However, melanin concentration in the retinal pigment epithelium (RPE) of black and white subjects has been reported to be similar, suggesting the RPE should not significantly affect the amount of light reaching the choroid between the two groups [60].

Our black subjects had larger FAZ sizes in the SCP and DCP than white subjects. It has previously been shown that the FAZ of healthy black individuals have larger foveal pit depth and larger pit diameter compared to white individuals [20]. The FAZ has been studied as a site where anatomic changes occur in the course of retinal pathologies, including DM with and without DR or DME [20,35,37,42,43,46,59]. The FAZ has been shown to enlarge with increasing severity of DR and DME [35,56]. Larger FAZ in young healthy adult black subjects may also contribute to higher susceptibility to ischemia in the fovea and diabetic maculopathy [58]. The underlying pathophysiology for the observed differences in macular capillary vasculature and FAZ features in the subjects in our study is unknown. An animal study found that pigmentation of melanocytes was inversely correlated with angiogenic activity of endothelial cells, suggesting that melanin content can affect downstream susceptibility profiles to angiogenic conditions [62]. Thus, the role of RPE and choroidal melanin content on the vasculature of the posterior segment should be investigated in future studies.

Limitations of this study include a relatively small sample size, which prevents the broad clinical applicability of our findings. Although our groups were well-matched for known factors that affect OCTA measurements, they represent a homogenously young and healthy adult sample of the population that is not typically afflicted with DM or DR. Thus, the microvasculature of older subjects with common systemic diseases should be examined [8,30,56]. Although we did not measure axial length of subjects, groups were well matched for refractive error with an equally small number of eyes with high myopia in each group [63,64]. Technological limitations included projection artifacts from the SCP onto the DCP, but we consistently obtained images of adequate quality for all subjects with our SSI cutoff. Correlation studies of retinal thickness and VD were not performed in this study because our OCTA AngioAnalytics software calculated the full retinal thickness of the imaged area, not the inner retinal layer thickness which is supplied by the retinal capillaries specifically analyzed by OCTA. Furthermore, total retinal thickness is not an accurate or ideal representation of the physiological makeup of the retina, which includes the distribution of Henle fibers, foveal cone packing, and integrity of the RPE and Bruch's membrane [63]. The automatically-detected measurements of retinal thickness, as obtained by the instrument we employed, do not account for the foveal morphology of our subjects' eyes. Therefore, using retinal thickness as a variable in a logistical regression analysis for vessel density would be misleading. The relationship between inner retinal layer thickness and VD should be investigated in future studies.

Additionally, our findings on the FAZ should be interpreted with caution because studies have suggested that OCTA segmentation of the FAZ into SCP and DCP is not meaningful because the vascular plexuses merge at the FAZ [21,46,65]. We could not perform a power analysis due to the lack of available, normative OCTA vascular data for healthy, young black and white adult populations upon which we could draw an estimated effect size. However, we believe our findings can serve as a starting point for future studies that investigate the establishment of normative OCTA databases. An important strength of our study is that our groups were well-matched for potentially confounding factors including age, gender, refractive error, and absence of medical and ocular conditions. With these variables being matched, our

method of comparing the OCTA values with linear mixed models without multivariate logistic regression was the appropriate statistical strategy which was confirmed by consultation with biostatisticians at our institution.

We acknowledge that our subjects' self-identified racial identities do not serve as an absolute proxy for ocular or skin melanin content and other physiological qualities, and that there are many other important external factors such as socioeconomic status, access to quality health care and medications, diet, exercise, and stress levels that can chronically influence the microvasculature of the retina and other internal organs [4,5,12,14,15,18,19,47,48,61,66]. However, these parameters are difficult to quantify in a consistent manner, and epidemiologic and genome-based studies on the differential risk of DR development in black populations have thus far had conflicting or indeterminate conclusions. Our study is, to our current knowledge, the first to quantitatively investigate the microvasculature in healthy young adult subjects in the US who differ by their self-identified race. In the face of current epidemiologic evidence of the racial disparities in DR, the function of racial identity in ocular health outcomes warrants investigation. Future investigations of the roles of melanin and the above-mentioned environmental factors on retinal vasculature could further clarify the underlying mechanisms for the reported racial disparities in DR prevalence.

## Conclusions

With these points in mind, our findings showing decreased blood flow in the DCP and in the choriocapillaris of our young and healthy adult black subjects compared to our white subjects may offer new insight into the reported differential vulnerabilities to DR. These findings could be significant in the context of the high prevalence of DR in black populations based on available epidemiological studies, and the current lack of consistent explanations that can account for this health disparity. Our findings suggest that people's racial or ethnic identity and other consequential factors should be accounted for in comparative research studies and in the development of normative databases for clinical practice. Accounting for potential differences in people's microvascular environment may improve guidelines for clinical management of retinal diseases. Future studies with larger and more diverse study groups are needed to better understand the microvascular environment of the retina, and to further elucidate any potential clinical associations among racial and ethnic identity, environmental and social factors, melanin content, macular capillary blood flow, and development of retinal disease.

## Supporting information

**S1 Table. Raw refractive data, OCTA (optical coherence tomography angiography) data, smoking history, and grouped demographic data of healthy young adult subjects analyzed in this study.** Each row depicts the spherical equivalent (SEq, as calculated as the sum of spherical error and half the value of cylindrical error), spherical error, cylindrical error, and OCTA Angioretina values of each eye that was imaged. OCTA Angioretina values include the superficial (spf) and deep vessel density in the whole 3x3mm macular image; area of the fovea; parafovea; superior and inferior hemifields of the 3x3mm macular image; the temporal, superior, nasal, and inferior quadrants; and the choroid blood flow area in the 3x3mm macular image, parafovea, and area of the fovea. The status of tobacco use and smoking, and packs per day of tobacco cigarettes, are also depicted. The table at the bottom of this data depict the following information for the 2 analyzed black and white groups: number of eyes, gender distribution, age, signal strength intensity (SSI), and spherical equivalent (SEq).
(XLSX)

## Author Contributions

**Conceptualization:** Dimitra Skondra.

**Data curation:** Lindsay Y. Chun, Megan R. Silas, Rose C. Dimitroyannis, Kimberly Ho, Dimitra Skondra.

**Formal analysis:** Lindsay Y. Chun.

**Funding acquisition:** Lindsay Y. Chun, Dimitra Skondra.

**Investigation:** Lindsay Y. Chun, Dimitra Skondra.

**Methodology:** Lindsay Y. Chun, Dimitra Skondra.

**Project administration:** Lindsay Y. Chun, Dimitra Skondra.

**Resources:** Lindsay Y. Chun, Dimitra Skondra.

**Software:** Lindsay Y. Chun, Dimitra Skondra.

**Supervision:** Dimitra Skondra.

**Validation:** Lindsay Y. Chun, Dimitra Skondra.

**Visualization:** Lindsay Y. Chun, Dimitra Skondra.

**Writing – original draft:** Lindsay Y. Chun, Dimitra Skondra.

**Writing – review & editing:** Lindsay Y. Chun, Dimitra Skondra.

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
