## [Decision Letter · Decision Letter 0]

23 Jul 2019

PONE-D-19-18713

Differences in macular capillary parameters between healthy black and white subjects with Optical Coherence Tomography Angiography (OCTA)

PLOS ONE

Dear Dr. Skondra,

Thank you for submitting your manuscript to PLOS ONE. After careful consideration, we feel that it has merit but does not fully meet PLOS ONE’s publication criteria as it currently stands. Therefore, we invite you to submit a revised version of the manuscript that addresses the points raised during the review process.

This is a novel study that all three reviewers found important and all 3 reviewers made excellent comments that will make this study better. We look forward to the revised version.

We would appreciate receiving your revised manuscript by Sep 06 2019 11:59PM. To enhance the reproducibility of your results, we recommend that if applicable you deposit your laboratory protocols in protocols.io, where a protocol can be assigned its own identifier (DOI) such that it can be cited independently in the future. For instructions see: http://journals.plos.org/plosone/s/submission-guidelines#loc-laboratory-protocols

We look forward to receiving your revised manuscript.

Kind regards,

Demetrios G. Vavvas

Academic Editor

PLOS ONE

Journal Requirements:

3. Please provide further details on how participants' race was classified. If participants selected their race from a list of possible options please include the list. If participants' were free to respond with free text please specify how responses were classified. Please ensure that you include this information in the Methods section.

4. Please ensure that your Methods section includes a detailed description of your  inclusion and exclusion criteria. Please ensure that you have specified which races were excluded from your study.

Reviewers' comments:

Reviewer's Responses to Questions

**Comments to the Author**

1. Is the manuscript technically sound, and do the data support the conclusions?

Reviewer #1: Yes

Reviewer #2: Yes

Reviewer #3: Yes

2. Has the statistical analysis been performed appropriately and rigorously? 

Reviewer #1: Yes

Reviewer #2: Yes

Reviewer #3: I Don't Know

3. Have the authors made all data underlying the findings in their manuscript fully available?

Reviewer #1: Yes

Reviewer #2: Yes

Reviewer #3: Yes

4. Is the manuscript presented in an intelligible fashion and written in standard English?

Reviewer #1: Yes

Reviewer #2: Yes

Reviewer #3: Yes

5. Review Comments to the Author

Reviewer #1: Interesting study informing retinal microvascular differences between healthy white and black individuals. It is good that the authors recruited young individuals and therefore showed that these differences are not part of the process of ageing but are rather present at baseline. However, please consider some comments below:

1. Did you match patients for blood pressure? A recent study reported decreased VD and increased FAZ in 3 × 3 mm macular scans of hypertensive patients compared to non-hypertensive controls (Lee et al., Sci Rep. 2019 Jan 17;9(1):156). Black individuals are more prone to hypertension and so hypertension could be a confounder in this analysis.

2. In the Participants section, mention what was the pool of candidate participants (e.g. patients visiting the clinic in a specific time interval?) and how you recruited the subjects. Also, did you perform any power analysis in order to determine the sample size?

3. In the phrase “prospective, observational cross-sectional study”, the word observational is redundant, since cross-sectional studies are always observational.

4. The phrase “Nonparametric Wilcoxon rank-sum (Mann-Whitney) tests were performed to match the racial groups for demographic and technology-related factors” is not valid. Please rephrase as following: “Nonparametric Wilcoxon rank-sum (Mann-Whitney) tests were performed to compare continuous demographic and technology-related variables between racial groups”.

Reviewer #2: -

general comments:

This is a well written paper that deserves publication. I think the authors did a great job in comparing the OCTA macula capillary parameters between white vs black subjects. The only paper that compared both groups used just OCT, not OCTA. The anatomic difference could potentially explain the vulnerability of black population to DR

specific comments

- Line 52. The authors concluded that healthy black subject have decreased macular capillary vasculature compared to whites and suggest that retina and choroidal vasculature difference may contribute to the vulnerability to DR. (1) I did not see any analysis of the choroid vasculature, but choriocapillaris in the paper, so perhaps the conclusion could be more specific (2) how about susceptibility to other vasculopathies like Lupus, Sickle cell etc?

- The OCTA figures demonstrating the difference between white vs black subjects. Pls provide legend and a better resolution images

- Line 116, The authors included “healthy young adults” and mentioned >18 y/o. It would be nice to clarify the range of age and a ref. for instance….young adult (ages 18-35 years; n = 97), middle-aged adults (ages 36-55 years, n = 197), and older adults (aged older than 55 years, n = 49).

- Based on the above range of ages, There was at least one middle aged adult when looking at the range of ages on the table, although minor observation, might be nice to clarify or add the reference with the range of ages that fits for this paper.

- Lines 51 Consider Adding always “adult” when mentioned young age. To read as “Young adults”.

- Line 72 add disease after “epidemic

- Line 242. consider changing “health” for anatomy. To read as “racial disparity in ocular anatomy”

- Line 323-… melanin concentration of the RPE of black and white subjects…. How about the choroid?

- Line 324 – please elaborate the possible role of melanin in vaculogenesis during development

-

Reviewer #3: The authors present a hypothesis geneating study, exploring the differences in capillary density between AA and Caucasian young individuals.They used an earlier version of he software and performed standard 2 layer segmentation to examine this question. the authors acknowledged the limitations of analysis and theitdataset, including the young age, inclusion of high myopia, the median myopic refraction of the population and the software. with that in mind, I have a few questions:

1. how did you correct for the inclusion of both eyes of each subject, which may skew the results?

2. I suggest you consider correcting for retina thickness, or at least exploring the relationship between thickness and VD. if the entire difference is driven by thinner retinas in black people, then they perhaps just have enough vasculature for their neuronal population.Also, we and others found that SSI can have a significant confounding effect on sVD, even when its in the >5 range, so please include that in your logistical regression as a confounder to adjust for. another possibility is also to adjust for refractive error, which also is a major confounder, these adjustments will address many of the limitations of the paper, even if they don't completely remove them.

Please include the statistical test used to arrive at the results in table 2, and all the confounders you actually adjusted for.

3. the foveal VD should be removed as it really only represents the known FAZ difference. by definition, fovea is avascular. central 1mm vd is largely reflecting the FAZ.

4. finally , the authors should acknowledge the fact that they did not adjust for multiple comparisons in their statistical analysis.

You might consider shortening the discussion and focusing all the theoretical oxygen discussions into one paragraph.

overall, a nice study which will hopefully generate discussion around this important topic.

6. PLOS authors have the option to publish the peer review history of their article (what does this mean?). If published, this will include your full peer review and any attached files.

Reviewer #1: No

Reviewer #2: Yes: Sandra R Montezuma

Reviewer #3: No

---

## [Author Response · Author response to Decision Letter 0]

6 Sep 2019

Please find our responses to each reviewer’s point below each comment.

Journal Requirements: 

We have made changes to the font and style of the manuscript as indicated by PLOS requirements. We have also named the files as requested as well.

We have included a caption for the Supporting Information at the end of the manuscript and updated in-text citations as appropriate.

3. Please provide further details on how participants' race was classified. If participants selected their race from a list of possible options please include the list. If participants' were free to respond with free text please specify how responses were classified. Please ensure that you include this information in the Methods section. 

Further details for this portion of the methodology have been included in the manuscript:

“All subjects provided demographic information including medical, ocular, and social history. Subjects self-identified their racial or ethnic identities by being asked to select from a list that included: black, white, East Asian, Southeast Asian, Indian, Native American, Latino, mixed, and other. Subjects who selected categories that were not solely “black” or “white,” or who selected multiple categories, were excluded from analysis due to low number in those group (fewer than 10 subjects per non-black and non-white groups). Subjects were also asked if they currently or had ever smoked or used tobacco products. Subjects who had never smoked or used tobacco products, or smoker fewer than 100 cigarettes in their life and currently did not smoke or use other tobacco products at the time of imaging, were categorized as not being a regular smoker and included in the study; others were excluded from this study (S1). Subjects who had a history of systemic medical conditions for which they required medications or interventions (such as diabetes, hypertension, hyperlipidemia, invasive or metastatic cancer, pulmonary conditions, coagulopathies) or ocular conditions (such as glaucoma, retinal disorders, uveitis, cataracts, blindness), and other conditions that could potentially affect the retinal microvasculature or prevent adequate imaging with OCTA, were excluded from this study. We determined that matching the subjects for blood pressure and other biomarkers of systemic health was not necessary, as we studied adult subjects in states of optimum health at baseline and prior to the onset of systemic conditions. Chart review was performed to confirm subjects’ available history.”

4. Please ensure that your Methods section includes a detailed description of your inclusion and exclusion criteria. Please ensure that you have specified which races were excluded from your study.

Further details for the inclusion and exclusion criteria have been included in the manuscript; please see above response to comment #3.

5. Review Comments to the Author

Reviewer #1: Interesting study informing retinal microvascular differences between healthy white and black individuals. It is good that the authors recruited young individuals and therefore showed that these differences are not part of the process of ageing but are rather present at baseline. However, please consider some comments below:

1. Did you match patients for blood pressure? A recent study reported decreased VD and increased FAZ in 3 × 3 mm macular scans of hypertensive patients compared to non-hypertensive controls (Lee et al., Sci Rep. 2019 Jan 17;9(1):156). Black individuals are more prone to hypertension and so hypertension could be a confounder in this analysis.

Thank you for this question. The subjects in this study were volunteers, mostly in their 20s, who had no history of blood pressure issues or concerns and were not and had never been on any medications for blood pressure-related issues. The study by Lee et al. that is referred to in this comment used the eyes of adult subjects with an average age range of 40 to 60.8 years, which is much different from that of our subjects.

We have modified our “Participants” section of Design and Methods to state: 

“We determined that matching the subjects for blood pressure and other biomarkers of systemic health was not necessary, as we studied adult subjects in states of optimum health at baseline and prior to the onset of systemic conditions.”

2. In the Participants section, mention what was the pool of candidate participants (e.g. patients visiting the clinic in a specific time interval?) and how you recruited the subjects. Also, did you perform any power analysis in order to determine the sample size?

“Subjects were clinic patients, students, and other members of the university and surrounding local community who were either approached and informed by study staff of this study, or who contacted study staff to participate after learning of the study independently via word-of-mouth during the time interval of this study.”

Regarding power analyses – Because normative databases for OCTA measurements of different populations have not yet been established, and because we report multiple OCTA measurements of our study groups, performing an a priori analysis for sample size was not appropriate. Performing a post-hoc analysis to determine the power of our study would be misleading; our statistically significant results would overestimate the power of our study, and our nonsignificant results would underestimate the power of our study, leading to contradictory conclusions (Please refer to Gelman and Carlin, 2014, DOI: 10.1177/1745691614551642). A retrospective post-hoc analysis would be difficult to perform, as there is a lack of currently available studies on OCTA findings of black and white young, healthy adult populations upon which we can draw an effect size.

We have added the following clarification in our “Limitations”:

“We could not perform a power analysis due to the lack of available, normative OCTA vascular data for healthy, young black and white adult populations upon which we could draw an estimated effect size. However, we believe our findings can serve as a starting point for future studies that investigate the establishment of normative OCTA databases.”

3. In the phrase “prospective, observational cross-sectional study”, the word observational is redundant, since cross-sectional studies are always observational.

The term “observational” has been removed.

4. The phrase Nonparametric Wilcoxon rank-sum (Mann-Whitney) tests were performed to match the racial groups for demographic and technology-related factors” is not valid. Please rephrase as following: “Nonparametric Wilcoxon rank-sum (Mann-Whitney) tests were performed to compare continuous demographic and technology-related variables between racial groups”.

Thank you. This has been revised.

Reviewer #2: -

general comments:

This is a well written paper that deserves publication. I think the authors did a great job in comparing the OCTA macula capillary parameters between white vs black subjects. The only paper that compared both groups used just OCT, not OCTA. The anatomic difference could potentially explain the vulnerability of black population to DR

specific comments

- Line 52. The authors concluded that healthy black subject have decreased macular capillary vasculature compared to whites and suggest that retina and choroidal vasculature difference may contribute to the vulnerability to DR. (1) I did not see any analysis of the choroid vasculature, but choriocapillaris in the paper, so perhaps the conclusion could be more specific (2) how about susceptibility to other vasculopathies like Lupus, Sickle cell etc?

The terminology regarding our OCTA measurements has been revised to accurately reflect analysis of the choriocapillaris.

We have included in our discussion the statement: “Based on our findings, variances in susceptibilities to other vasculopathies of the retina, such as lupus and sickle cell anemia, could also be based in functional differences in retinal vascularization. Further studies are needed to investigate these relationships.”

- The OCTA figures demonstrating the difference between white vs black subjects. Pls provide legend and a better resolution images

We have enhanced the images. Legend and captions for each figure are in the Manuscript. Thank you.

- Line 116, The authors included “healthy young adults” and mentioned >18 y/o. It would be nice to clarify the range of age and a ref. for instance….young adult (ages 18-35 years; n = 97), middle-aged adults (ages 36-55 years, n = 197), and older adults (aged older than 55 years, n = 49).

We have clarified our results section by adding: “23 subjects self-identified as black (age range: 22.4-32.4y, n=21; 41.1-47.5y, n=2) and 24 as white (age range: 22.3-32.4y, n=21, 40.3-48.0, n=3) (both groups non-Latino).”

- Based on the above range of ages, There was at least one middle aged adult when looking at the range of ages on the table, although minor observation, might be nice to clarify or add the reference with the range of ages that fits for this paper.

Because very few of our subjects in each group were of middle age, we have clarified by writing: “93 eyes of 47 mostly young, healthy, adult nonsmoking adult subjects were included in this study.”

- Lines 51 Consider Adding always “adult” when mentioned young age. To read as “Young adults”.

We have added “Adult” throughout the manuscript. Thank you.

- Line 72 add disease after “epidemic

We have added “disease” in the appropriate line.

- Line 242. consider changing “health” for anatomy. To read as “racial disparity in ocular anatomy”

We will maintain “health” in this particular phrase, as it better fits the overarching aim to determine what potential factors—anatomical, physiological, or others—play a role in the disparities in diabetic retinopathy.

- Line 323-… melanin concentration of the RPE of black and white subjects…. How about the choroid?

Above the statement regarding melanin in the RPE, we address the choroid as well: “However, the quantification of BFA in the choriocapillaris may have been affected by the larger amount of light absorption by the higher melanin content in the choroid of black subjects, so we cannot be certain that decreased choriocapillaris BFA seen in black subjects is representative of a true difference in the measured flow and not the result of artifacts related to different pigment content and light absorption.”

- Line 324 – please elaborate the possible role of melanin in vasculogenesis during development

We have clarified this point by adding in the “Discussion” section: “An animal study found that pigmentation of melanocytes was inversely correlated with angiogenic activity of endothelial cells, suggesting that melanin content can affect downstream susceptibility profiles to angiogenic conditions (62). Thus, the role of RPE and choroidal melanin content on the vasculature of the posterior segment should be investigated in future studies.”

Reviewer #3: The authors present a hypothesis generating study, exploring the differences in capillary density between AA and Caucasian young individuals. They used an earlier version of the software and performed standard 2 layer segmentation to examine this question. the authors acknowledged the limitations of analysis and their dataset, including the young age, inclusion of high myopia, the median myopic refraction of the population and the software. with that in mind, I have a few questions:

1. how did you correct for the inclusion of both eyes of each subject, which may skew the results?

Thank you for this question. 

We explained in our “Statistical analysis” section of Methods to state: “Linear mixed models were used to estimate the differences in retinal characteristics between the two groups. Linear mixed models were used to account for potential correlations between the left and right eyes of an individual.”

To make this clearer, we also added: “This method would correct for potential skewing of results due to the use of two eyes of most subjects.”

2. I suggest you consider correcting for retina thickness, or at least exploring the relationship between thickness and VD. if the entire difference is driven by thinner retinas in black people, then they perhaps just have enough vasculature for their neuronal population. Also, we and others found that SSI can have a significant confounding effect on sVD, even when its in the >5 range, so please include that in your logistical regression as a confounder to adjust for. another possibility is also to adjust for refractive error, which also is a major confounder, these adjustments will address many of the limitations of the paper, even if they don't completely remove them. Please include the statistical test used to arrive at the results in table 2, and all the confounders you actually adjusted for.

Thank you very much for these important comments.

Regarding retinal thickness – The vessel densities we report in this study represent the proportion of retinal tissue supplied by blood vessels. Thus, the vessel densities inherently represent the relationship between thickness and vascularization. The OCTA automatically detects the different retinal layers of each eye and determines the proportion of vascularized tissue per analyzed layer of retina—not an absolute count of vessels. Thus, the vessel density may be a more meaningful, physiological measurement as opposed to a solely anatomical one. 

In our Introduction, we refer to previous articles studying differences in retina morphology in different groups of people. We also refer to the critical point of these studies, which was that the morphology of the fovea—not just the retinal thickness— was a more meaningful, multifactorial component of retinal physiology. As seen in the work of Carroll et. al, the depth, diameter, and sloping of the foveal pit can all affect measurements of retinal thickness. Thickness alone is not an accurate or ideal representation of the physiological makeup of the retina, which includes the distribution of Henle fibers, foveal cone packing, and integrity of the RPE and Bruch’s membrane. The automatically-detected measurements of retinal thickness, as obtained by the instrument we employed, do not account for the foveal morphology of our subjects’ eyes. Therefore, using retinal thickness as a variable in a logistical regression analysis for vessel density would be misleading, especially in the face of previously established findings that suggest there is more to the retinal architecture to consider than thickness. These findings on foveal morphology and retinal anatomy do not contradict or diminish our findings, but rather show that further studies are required to bring different techniques together to create a cohesive scientific narrative. 

We have thus added a comment in our discussion as part of our limitations: “Correlation studies of retinal thickness and VD were not performed in this study because our OCTA AngioAnalytics software calculated the full retinal thickness of the imaged area, not the inner retinal layer thickness which is supplied by the retinal capillaries specifically analyzed by OCTA. Furthermore, total retinal thickness is not an accurate or ideal representation of the physiological makeup of the retina, which includes the distribution of Henle fibers, foveal cone packing, and integrity of the RPE and Bruch’s membrane (63). The automatically-detected measurements of retinal thickness, as obtained by the instrument we employed, do not account for the foveal morphology of our subjects’ eyes. Therefore, using retinal thickness as a variable in a logistical regression analysis for vessel density would be misleading. The relationship between inner retinal layer thickness and VD should be investigated in future studies.”

Regarding SSI – The average SSI of the black and white groups in our study was 70.93 and 71.13, respectively. That is a difference of 0.2%. There was also no significant difference in SSI between the two groups as shown in Table 1 (p=0.96), and thus using SSI as a variable in a logistic regression model of vessel density would be overcorrecting for SSI and produce inaccurate results. This was analyzed using the nonparametric Wilcoxon rank-sum (Mann-Whitney) test. Based on this statistical principle, we respectfully maintain that our original strategy for analysis is the most accurate. 

Regarding refractive error – As with the SSI, the average refractive error (in spherical equivalents) of our 2 groups was very similar with a 14% difference (p=0.41), and thus the potentially confounding effect of this variable was nullified. This was analyzed using the nonparametric Wilcoxon rank-sum (Mann-Whitney) test. As above, using refractive error as a variable in logistic regression would be overcorrection, which would be statistically inaccurate to do.

Regarding Table 2 – The statistical methods used to arrive at the results in Table 2 were fully described in the “Statistical analysis” section of our Methods. We formally consulted professional biostatisticians at the University of Chicago who helped us design the most statistically appropriate methods of analysis. As explained above, the appropriate potential confounders were corrected for via the nonparametric Wilcoxon rank-sum (Mann-Whitney) tests to ensure the 2 analyzed groups were not statistically different in terms of these confounding variables.

“These data are presented as means ± standard deviation. Linear mixed models were used to estimate the differences in retinal characteristics between racial groups. Linear mixed models were used to account for potential correlations between the left and right eyes of an individual. All retinal data are presented as means ± standard deviation unless specified otherwise (Table 2). A p-value of <0.05 was considered statistically significant.”

3. the foveal VD should be removed as it really only represents the known FAZ difference. by definition, fovea is avascular. central 1mm vd is largely reflecting the FAZ.

Thank you for your insightful and important comment. We agree that foveal VD data are affected by FAZ since it is different between the two groups. However, since foveal data for the VD is used automatically by the OCTA to calculate the VD of the 3x3mm circular macular image, we have kept the fovea data but in order to address this issue. 

We have added a comment that foveal VD and total 3x3mm VD data are likely affected by FAZ size: “Since the fovea is mostly avascular, any foveal differences may be best represented by the FAZ. However, due to the fact that foveal VD data is automatically used by the OCTA software to calculate the VD in the 3x3mm circular macula, we have included the foveal VD data.”

4. finally , the authors should acknowledge the fact that they did not adjust for multiple comparisons in their statistical analysis.

We have addressed this point by revising our “Limitations” section to state: 

“An important strength of our study is that our groups were well-matched for potentially confounding factors including age, gender, refractive error, and absence of medical and ocular conditions. With these variables being matched, our method of comparing the OCTA values with linear mixed models without multivariate logistic regression was the appropriate statistical strategy which was confirmed by consultation with biostatisticians at our institution .”

You might consider shortening the discussion and focusing all the theoretical oxygen discussions into one paragraph.

Thank you; we have condensed the paragraphs in the Discussion section regarding oxygen use and metabolism in the retina.

overall, a nice study which will hopefully generate discussion around this important topic.

---

## [Decision Letter · Decision Letter 1]

16 Sep 2019

Differences in macular capillary parameters between healthy black and white subjects with Optical Coherence Tomography Angiography (OCTA)

PONE-D-19-18713R1

Dear Dr. Skondra,

We are pleased to inform you that your manuscript has been judged scientifically suitable for publication and will be formally accepted for publication once it complies with all outstanding technical requirements.

With kind regards,

Demetrios G. Vavvas

Academic Editor

PLOS ONE

Additional Editor Comments (optional):

Reviewers' comments:

Reviewer's Responses to Questions

**Comments to the Author**

1. If the authors have adequately addressed your comments raised in a previous round of review and you feel that this manuscript is now acceptable for publication, you may indicate that here to bypass the “Comments to the Author” section, enter your conflict of interest statement in the “Confidential to Editor” section, and submit your "Accept" recommendation.

Reviewer #2: All comments have been addressed

Reviewer #3: All comments have been addressed

2. Is the manuscript technically sound, and do the data support the conclusions?

Reviewer #2: Yes

Reviewer #3: Yes

3. Has the statistical analysis been performed appropriately and rigorously? 

Reviewer #2: Yes

Reviewer #3: Yes

4. Have the authors made all data underlying the findings in their manuscript fully available?

Reviewer #2: Yes

Reviewer #3: Yes

5. Is the manuscript presented in an intelligible fashion and written in standard English?

Reviewer #2: Yes

Reviewer #3: Yes

6. Review Comments to the Author

Reviewer #2: the authors did a great job addressing all the comments. this is a well written paper and the statistical analysis been performed appropriately, very interesting paper. I agree to be accepted for publication

Reviewer #3: the authors have addressed my concerns adequately.

7. PLOS authors have the option to publish the peer review history of their article (what does this mean?). If published, this will include your full peer review and any attached files.

Reviewer #2: Yes: Sandra R Montezuma

Reviewer #3: No

---

## [Editor Report · Acceptance letter]

1 Oct 2019

PONE-D-19-18713R1 

Differences in macular capillary parameters between healthy black and white subjects with Optical Coherence Tomography Angiography (OCTA) 

Dear Dr. Skondra:

I am pleased to inform you that your manuscript has been deemed suitable for publication in PLOS ONE. Congratulations! Your manuscript is now with our production department. 

With kind regards,

on behalf of

Dr. Demetrios G. Vavvas 

Academic Editor

PLOS ONE